# *Cutibacterium acnes* Prosthetic Joint Infections: Is Rifampicin-Combination Therapy Beneficial?

**DOI:** 10.3390/antibiotics11121801

**Published:** 2022-12-11

**Authors:** Grégoire Saltiel, Vanina Meyssonnier, Younes Kerroumi, Beate Heym, Olivier Lidove, Simon Marmor, Valérie Zeller

**Affiliations:** 1Centre de Référence des Infections Ostéo-Articulaires Complexes, Groupe Hospitalier Diaconesses–Croix Saint-Simon, 125, rue d’Avron, 75020 Paris, France; 2Service de Médecine Interne et Infectiologie, Groupe Hospitalier Diaconesses–Croix Saint-Simon, 125, rue d’Avron, 75020 Paris, France; 3Laboratoire des Centres de Santé et Hôpitaux d’Île-de-France, Groupe Hospitalier Diaconesses–Croix Saint-Simon, 125, rue d’Avron, 75020 Paris, France; 4Service de Chirurgie Osseuse et Traumatologique, Groupe Hospitalier Diaconesses–Croix Saint-Simon, 125, rue d’Avron, 75020 Paris, France

**Keywords:** *Cutibacterium acnes*, prosthetic joint infection, rifampicin, exchange arthroplasty

## Abstract

No consensus has been reached on the optimal antibiotic regimen to treat *Cutibacterium acnes* PJIs (Ca-PJIs). In vitro studies showed excellent rifampicin efficacy against biofilm-associated *C. acnes* infections, but clinical studies did not confirm the superiority of rifampicin-combined therapy over monotherapy. This prospective cohort study was undertaken to analyze the outcomes of 70 patients who underwent exchange arthroplasty for chronic monomicrobial Ca-PJI and were treated with rifampicin or without between 2004 and 2019. The 37 patients treated from January 2004 to August 2014 were prescribed rifampicin-combination therapy and the 33 treated from September 2014 to December 2019 received monotherapy without rifampicin. The primary endpoint was the 2-year Kaplan–Meier-estimated reinfection-free probability, including relapses and new-pathogen PJIs. The 2-year reinfection-free rate was high and not different for patients who had received rifampicin or not (89.2% vs. 93.8%, respectively; *p* = 0.524). None of the patients relapsed and six developed new-pathogen PJIs. Our results do not support a benefit of rifampicin-combination therapy for patients who underwent exchange arthroplasty for chronic Ca-PJIs.

## 1. Introduction

*Cutibacterium acnes* [1] is a slow-growing Gram-positive anaerobic bacillus considered common skin flora with low pathogenicity. Its pathogenic role is now clearly established in various types of chronic infections, especially implant-associated infections [2,3,4,5,6,7], such as prosthetic joint infections (PJIs), infection following spinal fusion surgery, prosthetic valve endocarditis, neurosurgical shunt infections and endophthalmitis.

PJI is a devastating complication of joint-replacement surgery. The increasing number of joint replacements [8] has engendered an increased PJI incidence, with significant impact on patients’ morbidity and economic concerns [9]. Although *C. acnes* is involved in less than 10% of PJIs, it is the fourth most frequent species isolated from chronic PJIs [10,11] and the most frequent anaerobic bacterium found in PJIs. No hematogenously acquired infections have been observed [11,12,13]. *C. acnes* is by far the most frequent microorganism isolated from prosthetic shoulder infections, followed by *Staphylococcus epidermidis* [14]. It has been isolated from up to 50% of monomicrobial prosthetic shoulder infections. It is much rarer in prosthetic knee or hip infections [14]. Diagnosis can be challenging because of the infection’s very indolent course. Fever is generally absent, local inflammatory signs or a fistula are observed in less than one-third of the patients with these hip or knee infections and C-reactive protein (CRP) can be normal or only slightly elevated [15,16]. The most constant sign is persistent joint pain and functional disability. Preoperative joint aspiration, with differential blood cell count and prolonged cultures on enriched media, is key to confirming the diagnosis [15,16]. *C. acnes* is also found in polymicrobial PJIs.

No consensus has been reached on the optimal treatment of *C. acnes* PJIs (Ca-PJIs). Infectious Diseases Society of America guidelines [17] recommend first-line monotherapy, with penicillin G or ceftriaxone, and clindamycin or vancomycin as alternatives. According to French guidelines [18], no data have shown the value of combination therapy, especially with rifampicin. Moreover, the International Consensus on Orthopedic Infections [19] stated that the role of rifampicin-combination therapy remains unclear and that data on the benefit of rifampicin in PJIs are limited, even though *C. acnes* forms a robust biofilm on implant surfaces. *C. acnes* is highly susceptible to a wide range of antibiotics; however, the rates of clindamycin-resistant strains have increased recently [20,21]. Khassebaf et al. [22] reported that 9% of 55 *C. acnes* isolates were clindamycin-resistant. Rifampicin resistance has also been described [23].

Rifampicin efficacy against *C. acnes* biofilm has been studied in vitro and in an experimental foreign-body infection model; its efficacy was excellent, showing superiority over all other antibiotics [24]. However, three clinical studies on 128 patients with Ca-PJIs found no difference in success rates between patients treated with rifampicin or without [25,26,27]. More recently, based on a large clinical retrospective multicenter study including 187 patients with Ca-PJIs, Kusejko et al. [28] concluded that rifampicin-combination therapy was not markedly superior for the treatment of these infections. In light of the mixed findings previously reported and in their study, it remains inconclusive as to whether rifampicin should be recommended.

The aim of this cohort study was to analyze the outcomes of patients who underwent exchange arthroplasty for chronic monomicrobial Ca-PJIs and had been treated with rifampicin or without.

## 2. Results

### 2.1. Population

During the study period, among the 1838 PJIs (59% hip, 37.5% knee and 3.5% shoulder) managed in our Referral Center, 130 (7%) were Ca-PJIs. Sixty Ca-PJIs were excluded for the reasons specified under Methods and in Figure 1. For the seven patients who stopped rifampicin before 14 days, reasons for drug withdrawal were gastrointestinal intolerance for four, and one each had a skin rash, suspected staphylococcal PJI, or was not indicated.

Finally, 70 patients with chronic monomicrobial Ca-PJIs of the hip, knee, or shoulder, who underwent exchange arthroplasty, were included in the study. Thirty-seven patients received rifampicin-combination therapy and thirty-three were given monotherapy without rifampicin (Figure 1).

Patients’ characteristics on the day of reintervention for exchange arthroplasty, henceforth referred to as baseline, are reported in Table 1. Most Ca-PJIs were late chronic infections (80%); none were hematogenous. More patients with an American Society of Anesthesiology (ASA) score >2 were in the monotherapy group and more rifampicin-treated patients had ≥2 previous surgeries on the affected joint. All *C. acnes* strains were rifampicin-susceptible and three were clindamycin-resistant.

### 2.2. Antibiotics and Surgery

Modalities of antibiotic therapy and surgical interventions are detailed in Table 2.

The median [interquartile range (IQR)] duration of IV antibiotic administration was longer for rifampicin recipients (30 (28–42) days) than those not treated with it (19 (14–26) days) (*p* < 0.0001). Intravenous (IV) antibiotics were followed by oral intake for all but two patients. Rifampicin was continued orally for only five patients (Table 2).

The median [IQR] duration of total antibiotic therapy was also longer for rifampicin recipients (84 (84–91) days) than those not receiving rifampicin (43 (42–84) days) (*p* < 0.0001).

Most patients (*n* = 62, 89%) were managed with one-stage exchange arthroplasty. Among them, 29 received rifampicin-combination therapy and 33 did not. The eight patients who underwent two-stage exchange arthroplasty received rifampicin.

### 2.3. Outcomes

Outcomes are detailed in Table 2. Median [IQR] duration of follow-up was longer for rifampicin recipients (95 (71–125) months) than those not treated with rifampicin (36 (26–45) months) (*p* < 0.0001).

The Kaplan–Meier estimated 2-year reinfection-free probability did not differ between rifampicin recipients (89.2% [+/−0.05]) and patients not treated with rifampicin (93.8% [+/−0.04]) (*p* = 0.524) (Figure 2).

No relapse or PJI-related deaths occurred. Six patients developed new infections, all within 1 year, four rifampicin recipients and two not prescribed rifampicin. Details on these new pathogen infections are reported in Table 3. Three were classified as acute hematogenous infections. A Gram-positive cocci microorganism was isolated from five new PJIs. Isolates from one rifampicin recipient and one patient not treated with that antibiotic were rifampicin-resistant. Data were not available for one patient (Table 4).

Considering only patients who had undergone one-stage exchange arthroplasty, the reinfection-free rates did not differ (*p* = 0.603) between rifampicin recipients (96.6% [+/−0.03]) and for those not given rifampicin (93.8% [+/−0.04]).

### 2.4. Treatment-Related Adverse Events

Antibiotic-attributed adverse events, their type, grade and engendered treatment changes are shown in Table 5 and Table 6. Rates of adverse event-induced antibiotic withdrawal rates were comparable for patients treated with rifampicin or without.

## 3. Discussion

This observational, prospective cohort study investigated 70 patients with chronic Ca-PJIs managed with exchange arthroplasty and antibiotic regimens including rifampicin or not. None of them experienced relapses and the 2-year reinfection-free rates did not differ between groups. Our results do not support a benefit of rifampicin-combination therapy for patients treated with exchange arthroplasty for Ca-PJIs.

Use of rifampicin to treat Ca-PJIs is essentially based on experimental data, showing its excellent efficacy against *C. acnes* biofilm [24]. Although the major role of biofilm in implant-associated infections is well-established [29], a rifampicin-combination benefit has only been proven for *Staphylococcus aureus* PJIs managed with prosthesis retention [30,31]. Rifampicin is not recommended to treat streptococcal, enterococcal or for Ca-PJIs [17,18]. However, the question of the benefit of combination therapy with rifampicin remains [25,26,27,28]. It was recently analyzed by Kusejko et al. in a large retrospective multicenter study on the outcomes of *Cutibacterium* PJIs [28]; they observed suggestive, but not statistically significant, evidence of a beneficial effect of rifampicin adjunction to the antibiotic regimen. They concluded that it is still inconclusive as to whether rifampicin should be recommended and that a dedicated prospective multicenter study is needed to resolve that issue. Unlike our study, their patients underwent various types of curative surgeries, including debridement and irrigation with prosthesis retention (DAIR) for 18%. They noted that clinical success predominantly resulted from prosthesis removal or prosthesis exchange, a major point on which we want to focus, since Ca-PJIs are almost always chronic infections requiring removal of the infected implants with the biofilm on them.

Further arguments against rifampicin use are its poor tolerance [32,33] and important drug–drug interactions [34]. Frequent rifampicin adverse events were also observed in our study, with four patients excluded from our analysis because of early adverse events leading to its withdrawal before 14 days. In addition, combined rifampicin–clindamycin, a frequent choice to treat Ca-PJIs [15,25], is limited by notable drug–drug interactions, especially when taken orally [35,36]. Therefore, we do not recommend using oral rifampicin–clindamycin to treat severe infections, such as PJIs.

Our cohort was clinically and microbiologically homogeneous, and in terms of surgical strategy, as we included only monomicrobial Ca-PJIs managed with exchange arthroplasty. Our patients’ characteristics were similar to those of previous studies [25,26,27,28]: mostly males, median age ~70 years and few comorbidities other than diabetes. PJIs were mainly chronic post-operative infections. Susceptibility patterns of the *C. acnes* isolates were also similar, with <10% being clindamycin-resistant [22]. Unlike other studies, the prosthetic shoulder-infection rate was low (19%), easily explained by having no shoulder surgeon on our team before 2016.

Ca-PJI treatment usually has a good success rate (80–87%) [25,27,28]. Our results confirmed that finding with a high two-year reinfection-free rate of 91.3%, and no PJI relapses or related deaths. Only new-pathogen PJIs, i.e., with a different microorganism, were observed: more than half of those patients had been subjected to multiple surgeries on the joint. Rifampicin-resistant bacteria were isolated from only two patients. The type of antibiotic did not appear to have an impact on the resistance profile of the new-PJI microorganisms. Unlike our findings, Kusejko et al. [28] reported 15% proven and possible relapses, certainly reflecting the high percentage of DAIR-treated chronic PJIs, as discussed by those authors. Indeed, relapses were more frequent for patients treated with DAIR, which was identified as a risk factor with a hazard ratio of 2.19 in multivariate analysis. Our results and theirs underline again the importance of the choice of the surgical strategy according to the PJI type and infection duration. In our experience, one-stage exchange arthroplasty is the first-choice therapy to treat these chronic PJIs [37]. At present, complete removal of the implants with the biofilm seems to be the best strategy to treat chronic PJIs and is an additional argument not to use rifampicin in this setting.

The question remains whether rifampicin is useful in the case of acute infections with prosthesis retention. A prospective randomized trial is needed to address that specific question but is difficult to achieve because this situation is rare. A French, multicenter, prospective randomized trial (RIFACute) [38] is planned to try to determine whether rifampicin adjunction to the regimen for acute and chronic Ca-PJIs is beneficial.

The last important aspect of managing these infections is the choice of the most effective antibiotic regimen and its duration. No antibiotic-therapy consensus exists for *Cutibacterium* infections [16,20,28]. The bacterium is highly susceptible to a wide range of antibiotics: β-lactams, clindamycin, rifampicin, vancomycin, daptomycin and fluoroquinolones. However, increasing clindamycin-resistance was reported [22]. *Cutibacterium* is naturally resistant to metronidazole. Various antibiotic regimens are recommended: first-choice penicillin or ceftriaxone and vancomycin or clindamycin as alternatives by the American guidelines [17]; and amoxicillin, cefazolin or clindamycin by the French guidelines (18). In our cohort, clindamycin was by far the most frequently prescribed IV (70%) and oral (80%) antibiotic. Reasons to use clindamycin as the first-choice agent are its very good in vitro efficacy (MIC 0.003–0.2 mg/L) [20], good bioavailability and clinical tolerance, 30–50% bone diffusion [39], low cost and narrow spectrum. However, clindamycin-resistance is not rare (4% herein) and must be verified before prescribing clindamycin. Amoxicillin is an alternative-of-interest and the other first-choice therapy in our experience. Indeed, its MICs are very low (0.028–0.117 mg/L) [20], with 80% bioavailability, good tolerance and also low cost; however, its bone diffusion is lower (10%) [40] than that of clindamycin. In the large retrospective study by Kusejko et al. [28], clindamycin and amoxicillin were the most frequently given mono- and combination therapies to treat Ca-PJIs, with ciprofloxacin and levofloxacin prescribed in combination with rifampicin. Renz et al. [16] also treated orthopedic *Cutibacterium* spp. implant-associated infections with levofloxacin–rifampicin combination therapy, which was associated with high rates of adverse events and treatment discontinuation [41], questioning the indication of such a complex regimen, when other effective and simpler alternatives are available. No published randomized or comparative trial has addressed that topic.

Antibiotic-therapy duration for PJIs usually ranges from 6 to 12 weeks. According to by Kusejko et al.’s time to-event-analysis [28], antibiotic administration over 6 weeks had a favorable impact on outcomes (hazard ratio 0.29; *p* = 0.0002). Treatment duration for our cohort’s monotherapy group was significantly shorter (43 vs. 84 days, *p* < 0.0001) because we shortened it from 12 to 6 weeks in 2017 for patients managed with exchange arthroplasty. As noted above, rifampicin recipients were included earlier, from January 2004 to August 2014, and those not receiving rifampicin from September 2014 to December 2019. Despite the shorter treatment duration and no rifampicin, monotherapy-group patients did not experience relapses and their outcomes did not differ from those treated with rifampicin. These findings lead us to recommend 6-week antibiotic duration for Ca-PJIs managed with exchange arthroplasty.

Several limitations of our study have to be underlined. First, it was a monocenter study conducted in a highly specialized referral center for the treatment of complex BJIs, thereby limiting the generalizability of our findings and carrying a risk for selection bias. Second, the most important limitation is the existence of two different treatment periods, with or without rifampicin. Again, combined-rifampicin and monotherapy recipients were recruited at different periods. Treatments have evolved over time and could influence patients’ outcomes. Indeed, all patients who had undergone two-stage exchange arthroplasty were in the rifampicin group, and their treatment duration was significantly longer. Notably, no relapses occurred in either group, which is a strong argument supporting the fact that rifampicin does not provide additional benefit to an antibiotic regimen when implants are changed. It should be also underlined that patients treated without rifampicin had a shorter follow-up duration than those having received rifampicin (36 vs. 95 months). We cannot exclude late *C. acnes* relapses in this group that appeared after the study period. However, the duration of follow-up in this group was at least two-years with a median of 3 years, which is a quite appropriate follow-up time and is longer than those reported by other studies [25,26,27,28].

Other limitations of our study have to be mentioned. We included patients with different PJI sites (knee, hip or shoulder) that do not always have the same prognoses, but the distributions between patients treated with rifampicin or without did not differ significantly. Rifampicin use was defined as administration for at least 2 weeks and not during the entire treatment period. This may have underestimated the impact of rifampicin, but most patients received longer rifampicin treatments. Indeed, the median duration was 28 days. Concomitant use of rifampicin and clindamycin may have resulted in underdosing of clindamycin, but this risk was minimized by the use of high-dose continuous IV clindamycin infusion with systematic clindamycin drug-monitoring and dosage adjustment. Finally, although different treatment regimens were administered with a risk of confounding bias, most patients received clindamycin or cefazolin during the initial IV therapy, and clindamycin was by far the most frequent drug used (80% of the patients) during oral therapy.

## 4. Materials and Methods

### 4.1. Study Population

This cohort study was conducted in a French National Referral Center for Bone-and-Joint Infections [42]. All patients admitted to our Referral Center for PJIs are registered in the prospective PJI cohort (NCT 01963520, NCT 02801253). Each patient’s demographic, epidemiological, clinical, microbiological, therapeutic (surgeries and antibiotics), adverse event, and outcome information is entered prospectively. The primary outcome was the 2-year reinfection-free rate.

All patients, ≥18 years old, managed from January 2004 to December 2019 for a chronic (i.e., lasting >1 month) monomicrobial Ca-PJI, and treated with one- or two-stage exchange arthroplasty and antibiotics were included. Polymicrobial Ca-PJIs or patients operated on for aseptic loosening with positive intraoperative cultures were excluded.

Ca-PJI was defined as *C. acnes* isolation from ≥2 cultures of preoperative joint-fluid and/or intraoperative tissue specimens plus one or more of the following criteria: a sinus tract communicating with the prosthesis, local inflammatory signs (swelling, warmth, erythema), CRP > 5 mg/L and/or radiological parameters (i.e., periosteal bone formation, subchondral osteolysis) of infection [43,44].

*C. acnes* identification relied on results of intraoperative sample and/or preoperative joint-fluid–aspirate cultures, handled as previously described [12,45]. Joint aspiration was done in the Department of Radiology under fluoroscopic guidance and strict sterile condition. Absence of antibiotic therapy was verified before aspiration. After joint-fluid aspiration, saline was injected into the joint and then recovered. When possible, 2 samples were obtained after saline injection. Specimens were transported within 2 h to the Microbiology Laboratory, where differential white blood cells counts were determined by light microscopy. Synovial fluid was inoculated into PolyViteX chocolate agar (PVX, bioMérieux, Marcy-l’Étoile, France) (incubated under 5% CO_2_) and anaerobic Columbia agar plates (bioMérieux, Marcy-l’Étoile, France) and into aerobic (Hemoline, bioMérieux, Marcy-l’Étoile, France) and into anaerobic enrichment broths (Schaedler broth, bioMérieux, Marcy-l’Étoile, France). Aerobic and anaerobic cultures were incubated for 10 days until December 2015, and incubation for anaerobic cultures has since been prolonged to 14 days to optimize isolation of slow-growing bacteria. On day 10/14 or earlier, if bacterial growth was visible, broths were subcultured on PVX chocolate agar and anaerobic Columbia agar plates, and incubated at 37 °C for 48 h. Intraoperative samples were realized after standardized preoperative hygiene procedures. Before starting antibiotics, ≥3 intraoperative samples of bone and/or synovium that appeared inflamed or infected were obtained during surgery. They were processed aseptically within 2 h in a class-2 laminar air-flow safety hood. Specimens were disrupted by vigorous crushing in sterile mortars with sterile diluents. Aliquots of the resulting suspensions were cultured as described above for joint-fluid aspirates. Bacteria were identified to species with the rapid ID 32 API system (bioMérieux, Marcy l’Étoile, France) and, since 2012, by mass spectrometry (MALDI biotyper, Bruker Daltonik GmbH, Bremen, Germany). Antibiotic susceptibilities were determined with the standard disk-diffusion method, according to the recommendations of French Society of Microbiology (CaSFM) and the European Committee on Antimicrobial Susceptibility Testing (EUCAST) [46].

PJIs were classified according to three clinical settings previously described [12], based on initial PJI signs and derived from Tsukayama’s classification [47]: early postoperative PJI (onset of symptoms within 30 days after joint surgery), late chronic PJI (progressive symptoms occurring ≥30 days after surgery) and acute hematogenous infection, defined as sudden onset of local and general symptoms occurring after a symptom-free interval of ≥30 days post-surgery with identification of a portal of entry or recent bacteremia.

PJI-symptom duration was defined as the time between symptom onset and exchange arthroplasty in our Center.

### 4.2. Medical and Surgical Treatments

The medical–surgical management strategy was decided during multidisciplinary consensus meetings, guided by the patients’ comorbidities, surgical risk, and anatomical and functional status of the infected joint.

One-stage exchange arthroplasty was the first-choice strategy. Two-stage exchanges were done before January 2012 in a few patients who had undergone multiple (≥3) previous operations on the joint or if a large bone graft was required. Since 2012, no two-stage arthroplasty for monomicrobial Ca-PJI has been done because outcomes of patients who underwent one-stage exchanges were very good in our experience [37] and because of high antibiotic susceptibility of the isolated microorganism. Antibiotic-loaded bone cement was not used, in either the spacer or to fix the prosthesis.

For Ca-PJIs, initial IV clindamycin or cefazolin was administered. Vancomycin was given during the first week to 16 multi-operated patients awaiting the results of intraoperative sample cultures. Cefazolin, clindamycin and vancomycin were always administered by continuous IV infusion and serum-antibiotic levels were monitored, as described previously [35,48,49].

The antibiotic regimen was started during surgery after tissue sampling. Total antibiotic therapy lasted 12 weeks from 2004 to 2016. Since 2017, it has been shortened to 6 weeks, except for patients at high risk of relapse (≥3 operations, receiving chemotherapy or immunosuppressant(s), CHILD B or C cirrhosis, sickle-cell anemia, irradiated bone or requiring large bone grafts). Antibiotics were administered IV for 2–6 weeks, followed by an oral regimen to complete 6–12 weeks of therapy.

For the patients undergoing two-stage exchange arthroplasty, rifampicin was only used after the first stage with prosthesis removal. It was given postoperatively for 12 weeks. For the second-stage operation, performed one month after stopping antibiotics, patients received an antibiotic regiment (usually cefazolin) without rifampicin for 2 weeks, while waiting for cultures of intraoperative samples. If these remained sterile, antibiotic therapy was stopped.

According to our Center’s procedures, rifampicin-combination therapy was given systematically to treat Ca-PJIs until September 2014, but not thereafter for monomicrobial Ca-PJIs. Hence, rifampicin-treated patients had been included, from January 2004 to August 2014, and those not receiving rifampicin from September 2014 to December 2019 (Figure 1). To precisely analyze the benefit of rifampicin-combined therapy, the rifampicin-combination therapy group included only patients who received at least 14 days of rifampicin. Seven patients with <14 days of rifampicin administration were not included. Reasons for rifampicin withdrawal are listed in the Results.

The rifampicin dose was 600 mg twice a day, except for patients weighing >100 kg, who received 900 mg twice a day, and those weighing <50 kg, who received 450 mg twice a day. Rifampicin was administered IV initially then orally on an empty stomach, 1 h before a meal.

Treatment-associated adverse event severity was assessed according to the Common Terminology Criteria for Adverse Events (CTCAE) [50]. Antibiotic withdrawal was decided by the treating physician when severe events occurred (≥CTCAE grade 2).

### 4.3. Outcome Measures

All patients were monitored for at least 2 years. The following events were recorded: reinfection, either relapse with the same pathogen or new infection with a different microorganism, and death from any cause. The only patient lost-to-follow-up before 2 years had no sign of reinfection at the last follow-up visit at 10 months. This patient was excluded from the survival analysis.

All patients were seen as outpatients at the end of antibiotic therapy and then at 3 months, 6 months, 1 year, 2 years post-operatively, and then every 2 years. Patients, who did not attend their follow-up visits, were contacted by phone to collect data. At each visit, clinical (pain, fever, local inflammation) and radiological (appearance of periosteal bone apposition/radiolucent line, geodes...) signs of PJI were sought.

### 4.4. Statistical Analyses

The primary endpoint was the Kaplan–Meier-estimated cumulative reinfection-free probability at 2-year follow-up of the patients treated with rifampicin or without; they are expressed as percent (standard derivation). The secondary endpoint was the rate of antibiotic-associated adverse events for the patients treated with rifampicin or without.

All data were analyzed using SPSS version 25. Descriptive statistics are expressed as the number (percent) or the median [interquartile range (IQR)]. The Shapiro–Wilk method was applied to test data distribution. For bivariate analyses of continuous variables, Student’s *t*-tests were used for data with a normal Gaussian distribution; otherwise, the Mann–Whitney *U*-test was used. The frequency distributions of categorical variables for the two groups were compared with the χ^2^ test or the Fisher’s exact test, as appropriate according to the expected cell frequency.

### 4.5. Ethics Statement

All patients gave their written informed consent. The cohort was approved by the Île-de-France Ethics Committee.

## 5. Conclusions

The results of our prospective cohort study on chronic Ca-PJIs treated with exchange arthroplasty do not support rifampicin-in-combination to treat these infections. Our findings require confirmation by a multicenter randomized trial to definitively answer that question with a high level of evidence. The other important issue to be resolved is the optimal antibiotic therapy to treat these Ca-PJIs. Clindamycin, for susceptible strains, and amoxicillin seem to be the first-choice antibiotic monotherapies. The superiority of one over the other remains to be proven. The role of levofloxacin, alone or in combination, requires further investigation. It might be contributory for the treatment of polymicrobial Ca-PJIs, especially when staphylococci are present.

## Figures and Tables

**Figure 1 antibiotics-11-01801-f001:**
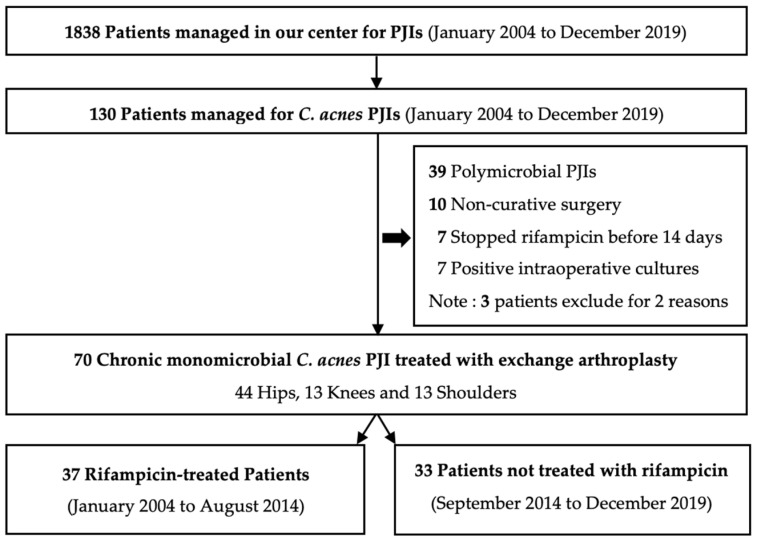
Ca-PJIs, *Cutibacterium acnes* prosthetic joint infections.

**Figure 2 antibiotics-11-01801-f002:**
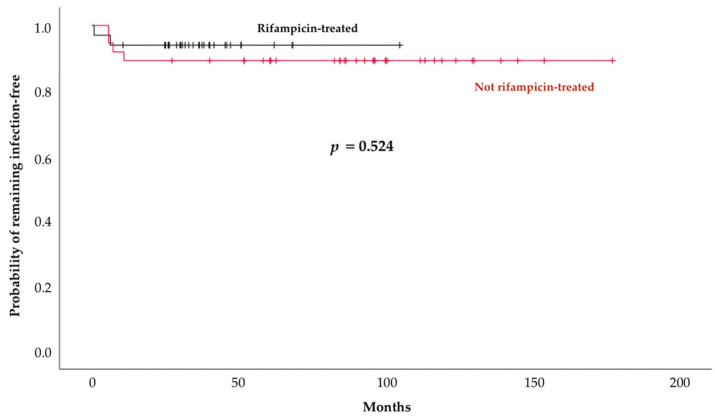
Kaplan–Meier estimated reinfection (relapse or new infection)-free probabilities for the 37 rifampicin recipients and the 33 patients treated without it for chronic Ca-PJIs.

**Table 1 antibiotics-11-01801-t001:** Baseline clinical characteristics of 70 patients treated for chronic *Cutibacterium acnes* PJIs with rifampicin or without.

	All	Rifampicin-Treated	Not Rifampicin-Treated	*p*-Value
Characteristics	*n* = 70	*n* = 37	*n* = 33	
Age, years, median [IQR]	69 (62–76)	70 (59–77)	69 (66–76)	0.937
Male, *n* (%)	50 (71)	27 (73)	23 (70)	0.796
Female, *n* (%)	20 (29)	10 (27)	10 (30)	0.796
Body mass index, kg/m^2^, median [IQR]	25 (23–29)	25 (22–29)	26 (23–29)	0.676
ASA score >2, *n* (%)	22 (31)	7 (19)	15 (45)	0.022
**Comorbidities**, *n* (%)				
Immunosuppressive treatment	2 (3)	0	2 (6)	0.219
Active neoplasia	3 (4)	3 (8)	0	0.242
Diabetes mellitus	10 (14)	4 (11)	6 (18)	0.499
Renal insufficiency (CrCl <60 mL/min)	2 (3)	1 (3)	1 (3)	1.000
**PJI characteristics**, *n* (%)				
Hip	44 (63)	23 (62)	21 (64)	1.000
Knee	13 (19)	10 (27)	3 (9)	0.069
Shoulder	13 (19)	4 (11)	9 (27)	0.123
Initial classification, *n* (%)				
Early post-operative	6 (9)	4 (11)	2 (6)	0.677
Late chronic	56 (80)	27 (73)	29 (88)	0.144
Not determined	8 (11)	6 (16)	2 (6)	0.266
Prior surgeries, *n* (%)				
1	39 (56)	16 (43)	23 (70)	0.032
≥2	31 (44)	21 (57)	10 (30)	0.032
Previous on-joint PJI, *n* (%)	6 (9)	4 (11)	2 (6)	0.677
Symptom duration before admission to our center, months, median (IQR)	12 (5–30)	14 (5–36)	12 (5–26)	0.824

PJI: prosthetic joint infection; IQR: interquartile range; ASA score: American Society of Anesthesiologists score; ClCr: creatinine clearance.

**Table 2 antibiotics-11-01801-t002:** Treatments and outcomes of patients treated for chronic *C. acnes* PJIs with rifampicin or without.

	All	Rifampicin- Treated	Not Rifampicin-Treated	*p*-Value
Treatments and Outcomes	*n* = 70	*n* = 37	*n* = 33	
**Antibiotic therapy**				
IV administration *				
Rifampicin, *n* (%)	37 (53)	37 (100)	—	NA
Duration, days, median [IQR]	28 (25–41)	28 (25–41)	—	NA
Clindamycin, *n* (%)	49 (70)	19 (51)	30 (91)	<0.0001
Duration, days, median [IQR]	20 (14–28)	28 (26–30)	14 (11–21)	0.583
Cefazolin, *n* (%)	34 (49)	21 (57)	13 (39)	0.161
Duration, days, median [IQR]	29 (10–42)	40 (29–43)	7 (6–13)	<0.0001
Vancomycin, *n* (%)	16 (23)	3 (8)	13 (39)	0.003
Duration, days, median [IQR]	7 (6–11)	6 (4–9)	7 (6–11)	0.306
Oral intake *				
Rifampicin, *n* (%)	5 (7)	5 (14)	—	NA
Duration, days, median [IQR]	42 (42–63)	42 (42–63)	—	NA
Clindamycin, *n* (%)	56 (80)	29 (78)	27 (82)	0.772
Amoxicillin, *n* (%)	6 (9)	1 (3)	5 (15)	0.193
Cefalexin, *n* (%)	6 (9)	5 (14)	1 (3)	0.203
None, *n* (%)	2 (3)	2 (5)	0	NA
Total duration				
Antibiotics, days, median [IQR]	84 (43–85)	84 (84–91)	43 (42–84)	<0.0001
IV antibiotics, days, median [IQR]	28 (19–34)	30 (28–42)	19 (14–26)	<0.0001
**Surgery**, *n* (%)				
1-stage replacement	62 (89)	29 (78)	33 (100)	0.006
2-stage replacement	8 (11)	8 (22)	0	0.006
**Outcomes**				
Follow-up duration, months, median [IQR]	60 (35–99)	95 (71–125)	36 (26–45)	<0.0001
Patients lost-to-follow-up <24 months, *n* (%)	1 (1)	0	1 (3)	NA
Reinfections, *n* (%)	6 (9)	4 (11)	2 (6)	0.677
Relapses	0	0	0	NA
New infections	6 (9)	4 (11)	2 (6)	0.677
After 1-stage replacement	3 (4)	1 (3)	2 (6)	0.599
After 2-stage replacement	3/8 (38)	3/8 (38)	—	NA

* Patients could successively receive vancomycin then cefazolin and clindamycin (or both alternately), based on microbiological results or in the case of intolerance. PJI: prosthetic joint infection; IQR: interquartile range.

**Table 3 antibiotics-11-01801-t003:** Characteristics of the six patients with a new PJI after treatment of *C. acnes* PJIs with rifampicin or without.

	PJI History and Comorbidities				*C. acnes* PJI				New PJI		
	PJI Site	Prior Surgery	Previous PJI	Comorbidities	Ca-PJI Classification	Last-Clean-To-Curative-Surgery Interval (Months)	PJI-Symptom Duration (Months)	Surgical & Antibiotic Treatments	Time of Onset after *C. acnes* PJI Treatment (Months)	Microorganism PJI Classification	Surgical & Antibiotic Treatments
**Rifampicin-Treated**											
1	Knee	≥2	Acute hematogenous *S. pneumoniae* PJI, 6 months earlier Treatment: DAIR and antibiotic therapy	None	Late chronic	6	6	1-stage exchange arthroplasty & IV: cefazolin + rifampicin (42 days) No oral treatment	12	Methicillin- susceptible *S. aureus* Acute hematogenous PJI	2-stage exchange arthroplasty & Antibiotics
2	Hip	1	None	None	Unknown	249	24	2-stage exchange arthroplasty & IV: cefazolin + rifampicin (42 days) Oral: cefalexin + rifampicin (42 days)	6	*Citrobacter freundii* Acute hematogenous PJI	DAIR & Antibiotics
3	Hip	≥2	Late chronic *Cutibacterium avidum* PJI, 7 years earlier Treatment: 1-stage exchange arthroplasty & antibiotics	Active cancer	Late chronic	81	40	2-stage exchange arthroplasty & IV: cefazolin + rifampicin (42 days) Oral: clindamycin (41 days)	8	Methicillin-susceptible *Staphylococcus capitis* Late chronic PJI	Suppressive antibiotherapy
4	Hip	≥2	None	None	Late chronic	6	3	2-stage exchange arthroplasty & IV: cefazolin + rifampicin (47 days) Oral: cefalexin + rifampicin (74 days)	7	Methicillin-resistant *Staphylococcus epidermidis* *Corynebacterium macginleyi* Positive intra-operative during new prosthesis implantation cultures	Antibiotics
**Not Rifampicin-Treated**											
5	Knee	1	None	None	Late chronic	17	12	1-stage exchange arthroplasty & IV: vancomycin followed by clindamycin (20 days) Oral: clindamycin (21 days)	7	*Streptococcus dysgalactiae* Acute hematogenous	DAIR & Antibiotics
6	Shoulder	≥2	None	None	Early post-operative	3	3	1-stage exchange arthroplasty & IV: vancomycin + clindamycin then cefazolin (14 days) Oral: amoxicillin (23 days)	1	Methicillin-resistant *Staphylococcus haemolyticus* Early post-operative	DAIR & Antibiotics

**Table 4 antibiotics-11-01801-t004:** Rifampicin, clindamycin and macrolide susceptibilities of bacterial strains isolated from new infections.

		Susceptibility		
Patient	New Infection Bacterial Strain	Rifampicin	Clindamycin	Macrolide
**Rifampicin-Treated**				
1	Methicillin susceptible *Staphylococcus aureus*	ND	ND	ND
2	*Citrobacter freundii*	/	/	/
3	Methicillin susceptible *Staphylococcus capitis*	Yes	Yes	Yes
4	Methicillin-resistant *Staphylococcus epidermidis* *Corynebacterium macginleyi*	No Yes	Yes Yes	Yes Yes
**Not Rifampicin Treated**				
5	*Streptococcus dysgalactiae*	Yes	Yes	Yes
6	Methicillin-resistant *Staphylococcus haemolyticus*	No	No	No

**Table 5 antibiotics-11-01801-t005:** Adverse events due to rifampicin, clindamycin, cefazolin or vancomycin.

	All	Rifampicin-Treated	Not Rifampicin-Treated	*p*-Value
Adverse Events	*n* = 70	*n* = 37	*n* = 33	
**Total, *n* (%)**	10 (14)	6 (16)	4 (12)	0.739
**Treatment discontinued, *n* (%)**	7 (10)	3 (8)	4 (12)	0.699
**According to antibiotic (IV or oral)**				
Rifampicin, *n*	37	37	*—*	
Adverse events, *n* (%)	5 (14)	5 (14)	*—*	NA
Vancomycin, *n*	16	3	13	
Adverse events, *n* (%)	1 (6)	1 (33)	0	0.003
Clindamycin, *n*	59	29	30	
Adverse events, *n* (%)	3 (5)	0	3 (10)	0.197
Cefazolin, *n*	34	21	13	
Adverse events, *n* (%)	1 (3)	0	1 (8)	0.161

**Table 6 antibiotics-11-01801-t006:** Antibiotic-related adverse events: type, grade and treatment changes made.

Antibiotic	Adverse Event, *n*	*n*	Treatment, *n*	
			Discontinued	Changed to
Rifampicin	Rash	1	1	Monotherapy without rifampicin
	Gastrointestinal disorders	4	1	
Vancomycin	Rash	1	1	Clindamycin
Clindamycin	Rash	2	2	Cefazolin or amoxicillin
	Gastrointestinal disorders	1	1	Intravenous clindamycin
Cefazolin	Liver toxicity	1	1	Clindamycin

All adverse events exceeded Common Terminology Criteria for Adverse Events grade 2.

## Data Availability

All the patients are registered in the prospective PJI cohort (NCT 01963520, NCT 02801253) database of our Referral Center for Bone and Joint Infections. The datasets used and/or analyzed during the current study are available from the corresponding author on reasonable request.

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
