# Peer review of "Cutibacterium acnes Prosthetic Joint Infections: Is Rifampicin-Combination Therapy Beneficial?"

_antibiotics, 2022, doi:10.3390/antibiotics11121801_

Round 1
Reviewer 1 Report
This is a well written manuscript with a clear message in a well-defined patient population. There are some limitations to the study like the relatively low sample size and the chance of selection bias, but this is well addressed by the authors.
I do not have many comments, some small suggestions/questions:
- How was rifampicin administered in the two-stage exchanges? During the implant free interval, afterwards, or both?
- There were no microbiological failures in both groups. For the no-rifa group this could be due to the fact that the follow-up was much shorter compared to the rifa-group, especially for Cutibacteria a follow-up of two years may be too short. This should be mentioned in the limitation section of the manuscript.
- Rifampicin use was defined as administration for at least 2 weeks. Previous papers demonstrate the importance of maintaining rifampicin throughout the whole treatment period for its benefit. This is a limitation that should be addressed in the discussion.
- Rifampicin has an interaction with clindamycin and lowers clindamycin serum levels. Some papers indicate that this is not a problem with clindamycin 600mg TID and rifampicin 450mg BID. However, in this paper the rifampicin dose is much higher. Could this be the reason that there is no benefit anymore? This might be good to mention in the discussion section of the paper.
Reviewer 2 Report
The authors submitted a very interesting manuscript on whether combined treatment with rifampicin has a beneficial effect in PJI.
In the present manuscript, some points are missing or unclear.
(1) In Table 1, the authors state that "50 male patients were included in the 70 cases," and what are the characteristics of the 20 female patients in the study?
(2) In the introduction (page 2, line 53), the authors state that the Infectious Diseases Society of America guidelines recommend first-line monotherapy with penicilin G, etc. Does this mean that the IDSA guidelines do not recommend combination treatment with rifampicin? Is there an explanation for why combination treatment with rifampicin is used in clinical therapy?
Reviewer 3 Report
The manuscript by Grégoire Saltiel et al. describes data on the benefits of a Rif combination in PJI caused by C. acnes.
As stated by the authors in the introduction, the manuscript is of little interest, as most of the information has already been demonstrated on several occasions.
In addition, modifications should be considered to obtain robust results.
Overall: italicize "in vitro," ".ie. ", " et al.", numbers less than twelve should be italicized.
The main problem is that all antibiotic treatments are very different from each other. This could be responsible for confounding bias. The authors should take this into account and analyze their cohort by stratifying their results, considering matched patients, or changing their selection criteria.
In addition, because the authors reported changes in their clinical practices in 2012 and 2017, they should verify that these changes did not significantly impact their results/conclusion.
Second, did the authors consider taking into account the bias of multiple statistical tests? Did they apply a correction to their results?
Finally, since the data used for these results could only be made available upon reasonable request, the authors should make them available to all readers by depositing them in a public database.
As a side note, which Reference Center for complex osteoarticular infections was involved? It seems to me that there are several in France. The culture methods must be detailed in more detail, because of the significant impact on the results obtained
Round 2
Reviewer 3 Report
My previous main comments were not taken into account and therefore I cannot recommend the publication.
"The main problem is that all antibiotic treatments are very different from each other. This could be responsible for confounding bias. The authors should take this into account and analyze their cohort by stratifying their results, considering matched patients, or changing their selection criteria."
This one was not addressed either:
"Point 4: In addition, because the authors reported changes in their clinical practices in 2012 and 2017, they should verify that these changes did not significantly impact their results/conclusion. Response 4: As suggested by the reviewer we procceded to compare outcome of the patients before and after 2012 and before and after 2017. There was no relapse whatever the periode of time. Regarding new infections, the comparison between the two periodes of time, prior (vs) after 2012 and 2017 did not show any differences."
This comment was not taken into account. I recommend a correction, as the same data was used in several statistical tests.
"Point 5: Second, did the authors consider taking into account the bias of multiple statistical tests? Did they apply a correction to their results? Response 5: This study is descriptive. No correction has been applied. We did not have multiple assumptions made in the methods, even none. The main objective mentioned in the method was re-infection free survival that was analysed by the Kaplan Meier test. Kh2/Fisher tests and Student/Mann Whitney U tests were used to compare while reporting descriptive data."
This issue could be resolved by anonymizing the data:
"Point 6: Finally, since the data used for these results could only be made available upon reasonable request, the authors should make them available to all readers by depositing them in a public database. Response 6: We are sorry, but French regulatory and ethics committees do not authorize deposition of personnal patient data in a public database."